# BALCONI: **BAL**ANCING **CON**TEXT AND INTERNAL KNOWLEDGE FOR TRAINING FLEXIBLE LLMS

## ABSTRACT

The faithfulness to the context is significant for large language models (LLMs) in tasks such as Retrieval-Augmented Generation (RAG) or Information Extraction. However, LLMs can exhibit a "stubborn" reliance on their internal knowledge, which leads to failure in maintaining faithfulness to the context. Ideally, a flexible model should leverage the given context if the user instruction requires to, yet remain correctness based on internal knowledge when the instruction does not provide the context. Considering such scenarios, we propose a balanced benchmark, FaithfulBench, to evaluate the faithfulness of LLMs, together with internal knowledge correctness in LLMs and evaluate whether the improvement in faithfulness would affect internal knowledge. Extensive experiments show that LLMs can be unfaithful to the context to some extent and in the Multi-choice QA, we observe an obvious negative correlation between faithfulness and internal knowledge correctness across different LLMs. Then based on the analysis of faithfulness enhancement methods, we find that instruction tuning using counterfactual data can significantly improve the model's context faithfulness, but compromise the model's internal knowledge. To address such a issue, we propose a straightforward yet effective approach BALCONI training by tuning with mixup data of factual requests, context requests, and NoAns (I cannot tell the answer from the context) requests. Experiments on our benchmark and a context-based machine translation task demonstrate that BALCONI can achieve a well-balanced effect in improving the balanced faithfulness and internal knowledge.

## 1 INTRODUCTION

Large Language Models (LLMs) have demonstrated remarkable capabilities in understanding and generating human-like text (Devlin et al., 2019; Bubeck et al., 2023), yet there remain various challenges in LLMs to output satisfactory response such as fairness (Gallegos et al., 2024), truthfulness (Kandpal et al., 2023) or faithfulness (Es et al., 2023). The faithfulness to the context, which refers to the model's ability to leverage context information to complete the task without relying on internal knowledge, is significance for LLMs in tasks such as RAG systems (Es et al., 2023), information extraction (Lu et al., 2022; Zhou & Chen, 2022), and summarization (Shi et al., 2023; Chen et al., 2022b). (Zhou et al., 2023). To this end, various methods have been to enhance the context faithfulness (Shi et al., 2023; Neeman et al., 2023; Zhou et al., 2023).

Ideally, the user should have control over the reliance on the context and internal knowledge, and a flexible model as an intelligent assistant (Figure 1) should leverage the given context if the user instruction requires to (such as explaining the meaning of 'can' in a sentence 'I don't have a can'), yet remain factual based on its internal knowledge when required an factual answer (such as close-book QA in querying what is the current president in USA). Furthermore, if the context does not contain the answer, the model should state 'I can't tell the answer from the context' (NoAns) (Zhou et al., 2023), when the instruction requires an answer based on the context, thus preventing potential user confusion. To our knowledge, no existing work has evaluated such the above flexibility systematically. While there has been work dedicated to context faithfulness (Zhou et al., 2023; Neeman et al., 2023; Li et al., 2023), they do not discuss how methods that increase context reliance could influence the use of internal knowledge.

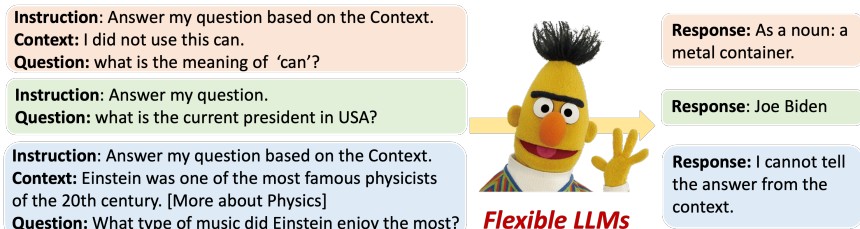

Figure 1: Examples for flexible LLMs in using context and internal knowledge.

To cover all three scenarios, we create FaithfulBench, which includes the tasks of OpenbookQA and Multi-choice QA based on NaturalQuestions (Kwiatkowski et al., 2019) and TriviaQA (Joshi et al., 2017) (Figure 2). We first evaluate the context faithfulness of LLMs on our benchmark, including the closed-source models `ChatGPT`, `GPT-4-Turbo`, and `Claude-3-Sonnet`, as well as the open-source models `Mistral-7B-Instruct` and `LLaMA-2-7B-Chat`. We observe a common trend, **all models exhibit a degree of stubbornness to their internal knowledge**, demonstrating challenges in adhering faithfully to the given context. Such finding are consistent with the existing work (Wu et al., 2024). Interestingly, despite the different capacity to output text-free answers in OpenbookQA, we observe **in Multi-choice QA there exists a significant negative correlation between faithfulness and internal knowledge correctness across different LLMs.** In fact, the robust memorization capabilities of these models may contribute to their stubbornness, as they tend to rely on their internal databases rather than adapting to external contextual information.

Second, we compare previous methods in improving faithfulness finding that the most effective method is instruction tuning using counterfactual data (Longpre et al., 2021), However, for the same question without a context, the correctness (measured by accuracy) of the tuned `Mistral-7B-Instruct` on internal knowledge reduces from 38.7% to 30.2%, and the model correctly outputs NoAns with a percent of only 2.3% when the context does not contain information to the questions. This phenomenon indicates that **tuning the model on counterfactual data can enhance model faithfulness to the relevant context but hinder the model's correctness in internal knowledge.** Furthermore, the model becomes prone to hallucinate rather than directly responding with NoAns when the context does not contain answers.

To address such limitations, we propose a straightforward yet effective approach, BALCONI training, tuning the model using mixup data of factual request, context requests, and NoAns requests. Our experiments demonstrate that BALCONI training using mixed data not only enhances the faithfulness of the model but also maintains a balanced performance across factual, context, and NoAns requests. Additionally, on out-of-distribution (OOD) tests, we find that BALCONI training also exhibits the most balanced performance among the scenarios in Figure 2. Finally for evaluation beyond QA tasks, experiments on context-based machine translation task further demonstrates the effectiveness of BALCONI. We will make our dataset and code available (**in supplementary files** during the review process). Our contribution can be summarized as follows:

1. We construct a benchmark including OpenbookQA and multi-choice QA to evaluate the faithfulness of LLMs to the context and corresponding internal knowledge.

2. We underscore the importance of balancing context faithfulness and internal knowledge, showing that fine-tuning exclusively on one aspect may inadvertently compromise the other.

3. We propose a novel training strategy BALCONI to strike a balance in context faithfulness and internal knowledge, validating the effectiveness in FaithfulBench together with a context-based machine translation task.

## 2 RELATED WORK

**Context Information.** Large language models (LLMs) exhibit remarkable capabilities, but they still encounter challenges when it comes to practical applications (Gao et al., 2023). These challenges include (1) hallucinations (Zhang et al., 2023; Lin et al., 2022); (2) slow knowledge updates (Lazaridou et al., 2021; Izacard et al., 2023); and (3) a lack of domain-specific knowledge (Kandpal et al., 2023). Some studies attempted to improve the model correctness by fine-tuning on factual

| OpenbookQA | Multi-Choice QA |
|---|---|
| **(i) *Context requests:*** **Instruction:** Answer my question using the context information, but not your own knowledge.
**Question:** Who had a 70s No 1 hit with Kiss You All Over?
**Context:** "Kiss You All Over" is a 1978 song performed by the **Elvis Presley**.... [MORE]
**Response: Elvis Presley** had a 70s No 1 hit with Kiss You All Over. | **(i) *Context requests:*** **Instruction:** Given the context information, select the correct option to the answer.
**Question:** Who had a 70s No 1 hit with Let's Do It Again?
**Option:** A. Staple Singers; B. The Beatles; C. The Rolling Stones; D. The Bee Gees; E. I can not answer the question with the context.
**Context:** "Let's Do It Again" was a hit song for the iconic British band **The Beatles**...[MORE]
**Response: B** |
| **(ii) *Factual requests:*** **Instruction:** Answer my question: Who had a 70s No 1 hit with Kiss You All Over?

**Response: Exile** had a 70s No 1 hit with Kiss You All Over. | **(ii) *Factual requests:*** **Instruction:** Answer the question with the option.
**Question:** Who had a 70s No 1 hit with Let's Do It Again?
**Option:** A. Staple Singers; B. The Beatles; C. The Rolling Stones D. The Bee Gees; E. I can not answer the question with the context.
**Response: A** |
| **(iii) *NoAns requests:*** **Instruction:** Answer my question using the context information, but not your own knowledge.
**Question:** Who had a 70s No 1 hit with Kiss You All Over?
**Context:** Merritt College ( born March 27, 1996) is an Italian - American actress .... [MORE]
**Response: I cannot tell the answer from the context.** | **(iii) *NoAns requests:*** **Instruction:** Given the context information, select the correct option to the answer.
**Question:** Who had a 70s No 1 hit with Let's Do It Again?
**Option:** A. Staple Singers; B. The Beatles; C. The Rolling Stones; D. The Bee Gees; E. I can not answer the question with the context.
**Context:** Merritt College ( born March 27, 1996) is an Italian-American actress .... [MORE]
**Response: E** |

Figure 2: Examples for flexible LLMs completing the OpenbookQA and Multi-choice QA tasks with contextual knowledge and internal knowledge.

knowledge (Neeman et al., 2023; Mecklenburg et al., 2024), but studies showed that fine-tuning on specific tasks would introduce forgetting in other capacity of LLMs (Luo et al., 2023). Leveraging context information could be an effective way to enhance the model performance such as reading comprehension, information extraction and retrieval-augmented generation (RAG). RAG dynamically retrieves information from external knowledge sources and uses the retrieved data as references to organize answers (Gao et al., 2023) which has been demonstrated to significantly reduce model hallucination and improve the correctness. However, studies have discovered that LLMs can overlook or ignore context (Kasai et al., 2024; Li et al., 2023; Si et al., 2022), which means the model is unfaithful to the context information. Some work have also studied how to respond to knowledge conflicts between the internal and context knowledge (Chen et al., 2022a; Neeman et al., 2023; Xu et al., 2024) or detect them (Wang et al., 2023). Differently, we focus on analyzing the balance problem of context faithfulness and internal knowledge correctness.

**Faithfulness of LLM.** Neeman et al. (2023) proposed a dataset to require the model to output disentangled contextual answer and internal answer at the same time, which focuses on handling knowledge conflict. Li et al. (2023) proposed that the knowledge should have a predefined prioritization that relevant context > models's parametric knowledge > irrelevant knowledge. But we focus on the practical scenario that the response should follow the user instructions without the specific prioritization. Zhou et al. (2023) proposed an opinion prompt in the third person perspective to enhance LLM's faithfulness without tuning the model. Wu et al. (2024) analyzed the faithfulness of LLMs and finding that the more the modified information deviates from the model's prior, the less likely the model is to prefer it. Different from previous studies, we study that the faithful model should reply the question by considering the user instructions and further analyse how the improvement of the faithfulness affects internal knowledge.

## 3 DATASET

We construct our FaithfulBench in two tasks, OpenbookQA, and Multi-choice QA. We form the task that given the data point $X_i = (Q_i, C_i, A_i, INST_i)$ where $Q_i$ is the question, $C_i$ is the context information, $INST_i$ is the instruction of the data and $A_i$ is the answer to the question, the faithful LLM should predict the correct answer $A_i$. The data contains three parts (Figure 2):

**(1) Factual requests** require the model to complete the instruction with the internal knowledge, where $C_i$ is empty, and $A_i$ is the factual answer to the question;

**(2) Context requests** require the model to complete the instruction using the information in the counterfactual context, where $C_i$ is the counterfactual context and $A_i$ is the counterfactual answer according to the context $C_i$.

**(3) NoAns requests** (I can't tell the answer from the context) require the model to complete the instruction with an unrelated context, where $C_i$ is the context unrelated to the question and $A_i$ is the answer 'I can't tell the answer from the context' (NoAns).

Factual requests mainly measure the model correctness of internal knowledge, and context and NoAns requests evaluate the model's faithfulness. To mitigate the models' reliance on a specific instruction template, we construct a instruction template pool for factual request and context requests. In the data of factual request, we randomly select one instruction from three templates. For context requests, we randomly select one template from a set of ten templates. The details of the templates are shown in the Appendix 7.11. For context requests, we follow (Zhou et al., 2023; Li et al., 2023) to construct counterfactuals to challenge the LLMs' faithfulness to the context, when facing knowledge conflicts.

## 3.1 OPENBOOKQA

We construct the OpenbookQA task based on the datasets Natural Questions (NQ) and TriviaQA. For NQ, we follow the framework of (Longpre et al., 2021), replacing the answer in the context with a counterfactual answer to create a counterfactual context and maintaining a consistent setting with previous work. Specifically, we adopt the method of corpus substitution and randomly select 2,000 data points as factual request and generate corresponding counterfactual contexts as context requests. Then, we randomly replace the context with ones that are unrelated to the question to create NoAns requests. This process yields 6,000 data points, including different types of requests. To evaluate the effectiveness of instruction tuning, we randomly split the dataset equally into a training set and an evaluation set.

For TriviaQA, which contains the Wiki passage questions and web search passages - we adopt the Wiki parts for in-distribution (ID) test. When making counterfactual data, we take special consideration to avoid conflicts in the context. For example in Figure 10, directly replacing a context with a counterfactual answer 'painting' still leaves information in the context implying the golden answer 'ballet' such as 'ballerina' or 'choreograoger'. The study (Xie et al., 2023) also points out that the context generated by directly replacing the answer is difficult for LLM to be convincing. The problem can be solved by using LLMs for content rewriting. In particular, we randomly select 2,000 data points from TriviaQA-Wiki as the factual request and create counterfactual contexts using `Claude-3-Sonnet` (the details of the prompt are shown in Appendix 7.2).

To control the data quality, we check whether the counterfactual answer and the factual answer exist in the counterfactual context, and regenerate the counterfactual until the former exists but the latter doesn't. 9% of the data are regenerated at least one time. After obtaining the data, we randomly select 100 instances and two PhD students in NLP area manually check whether the counterfactual contexts are satisfying, i.e. whether the counterfactual answer can be inferred from the context. Given the counterfactual context and the question with three options the factual one, the counterfactual one and NoAns, the annotators are required to select the option using the contextual information. We then calculate accuracies of them, which are 93% and 91%, respectively. The results indicate the quality of the LLM generated data is satisfactory. Then, we randomly replace the context with ones that are unrelated to the question to create the NoAns requests.

**Out-of-distribution (OOD) Test.** To further evaluate the fine-tuning methods to improve LLM's faithfulness, we also craft OOD tests based on the data from TriviaQA web search passages, as both NQ and Trivia-Wiki are based on Wiki passages. We randomly select 1,000 data points from TriviaQA-Web and generate context requests and NoAns requests in the same manner as the Trivia-Wiki part.

## 3.2 MULTI-CHOICE QA

As multi-choice QA is widely used to evaluate LLMs, we can directly challenge an LLM by presenting it with counterfactual answers, factual answers, and NoAns answers. An example of the multi-choice QA is shown on the right side of Figure 2. Given the question '*Who had a 70s No 1 hit with Let's Do it Again*' and five options, the factual answer is A., and according to the counterfactual context, the counterfactual answer is B. The model should output the correct options according to the instruction and context information. We construct the Multi-choice data based on the data in OpenbookQA. Besides the question and context (if provided), we present five options to the LLM,

including the counterfactual answer in the context request, the factual answer, the NoAns answer, and two additional incorrect answers, which are set in a random order. We generate the incorrect options by querying `Claude-3-Sonnet`, and the prompts are shown in the Appendix 7.4.

# 4 EXPERIMENTS

**Models.**  We evaluate the LLMs on the FaithfulBench, including the close-source models such as `ChatGPT` [1], `Claude-3-Sonnet`, and `GPT-4-Turbo` [2], and the open-source models such as `LLaMA-2-7B-chat`, `Mistral-7B-Instruct-v0.1` and the Mixture-of-Experts model `Mixtral-8x7B-Instruct-v0.1`. For the close-source models, we adopt the hyper-parameters *temperature=0.0* and *top_p=1.0* to mitigate the randomness of the outputs. For the close-source models, we use the greedy search for inference [3].

**Baselines.**  We also analyze the performance of previous faithfulness enhancement methods. (1) **Prompt**, directly requires the model to respond using the given prompts in our data; (2) **Decoding** (Shi et al., 2023), adopts a contrastive output distribution that amplifies the difference between the output probabilities when a model is used with and without context [4]; (3) **Attr**, (Zhou et al., 2023), designs the prompt that '*{Context} Question: {question} based on the given text? Answer:*'; (4) **Opin** (Zhou et al., 2023), designs the opinion-based prompt that '*Bob said, '{context}' Question: {question} in Bob's Opinion? Answer:*'; (5) **Opin-Inst** (Zhou et al., 2023), designs the prompt, '*Instruction: answer a question based on the provided input-output pairs. Bob said, '{context}'. Question: {question} in Bob's opinion? Answer:*'; (6) **SFT-C**, trains the model with the context requests to improve the model faithfulness, where the context is counterfactual (Longpre et al., 2021); (7) **SFT-NoAns**, trans the model with NoAns requests to enhance the model output 'NoAns' (Rajpurkar et al., 2018). For comparison, we implement **SFT-F**, that trains the model with the factual requests whose answers are the factual ones (Neeman et al., 2023).

**Implementations for Baselines.**  We adopt the model `Mistral-7B-Instruct-v0.1` (Jiang et al., 2023) for experiments comparing different methods. In the methods with supervised tuning, we train the model with QLoRA (Dettmers et al., 2024) on 8 GPU (Tesla A10 24G) using the Adam optimizer (Kingma & Ba, 2014), which can mitigate the computational cost. For all the methods, the batch size is 1 on each device, the gradient accumulation step is 8, the learning rate is 2e-5, and the scheduler is set cosine. The max sequence length of the instruction is 1400 and that of new tokens is set 200. Since response with a long answer are widely used in the recent LLMs and has widely applications, we extend the short answers in OpenbookQA to a sentence using `Claude-3-Sonnet` for the training (see Appendix 7.3). In each data split, the data are split to 1:1 for training and evaluation. We train our model 10 epochs and the final checkpoints are used for evaluation. For inference, we adopt greedy search for the reproduction of the results.

**Evaluation.**  For OpenbookQA, we first adopt the Match Score (MS) as a flexible measure of the inference of the LLMs, where we treat the answer that the short factual answer exists in the response as correct ones. For Multi-choice QA, we directly calculate the correct ratio of the selected options. Since it is difficult to directly output the options for 7B models without training, such as Prompt, Attr, Opin and Opin-Inst, thus, we calculate the log probability of the options and select the largest one as the prediction. For NoAns, we treat the response including ' not ' and 'I am sorry' as correct response for a flexible measure, since these responses mostly express cannot tell the answer. Note that after training the model, we evaluate the model on the context requests and NoAns requests of the test set but evaluate the model on the factual request in the training set for measuring the knowledge change after training. It is expected that the model context faithfulness could be enhanced without a sacrifice of their internal knowledge. We further propose a balanced faithfulness score (BFS) as a surrogate to evaluate the overall performance of faithfulness and internal knowledge correctness in the LLMs, which is the average performance of the context requests, factual request and NoAns requests.

---

[1] We adopt the `ChatGPT` version gpt-3.5-turbo-0125.

[2] We adopt the `GPT-4-Turbo` version gpt-4-0125-preview.

[3] In Multi-Choice QA, for the small models like `LLaMA-2-7B-Chat` and `Mistral-7B-Inst-v0.1`, we adopt the log probability on the options as the selected answers since the models tend to not output options. For larger models, we directly extract the options by regular expression.

[4] We set the hyperparameter following the github script, that WEIGHT = 2_-1 and TOPP=0.0.

Table 1: The performance of different LLMs on our FaithfulBench. The metric for OpenbookQA is match score and that for Multi-chocie QA is the accuracy of the options.

| Dataset | NaturalQuestion | | | | TriviaQA-Wiki | | | |
|---|---|---|---|---|---|---|---|---|
| Split | Context | Factual | NoAns | BFS | Context | Factual | NoAns | BFS |
| **OpenbookQA** | | | | | | | | |
| LLaMA-2-7B-chat | 58.5 | 43.1 | 50.5 | 50.7 | 56.6 | 55.9 | 18.5 | 43.7 |
| Mistral-7B-Inst-v0.1 | 63.2 | 38.7 | 21.3 | 41.1 | 61.1 | 43.1 | 13.3 | 39.2 |
| ChatGPT | 58.8 | 58.3 | 59.9 | 59.0 | 53.9 | 81.0 | 10.6 | 48.5 |
| Claude-3-Sonnet | 66.7 | 59.9 | **97.0** | **74.5** | **76.2** | 76.9 | **91.8** | 81.6 |
| Mistral-8x7B-Inst-v0.1 | **68.9** | 59.3 | 78.7 | 69.0 | 72.7 | 75.4 | 70.0 | 72.7 |
| GPT-4-Turbo | 60.5 | **61.9** | 33.4 | 51.9 | 45.2 | **83.9** | 12.0 | 47.0 |
| **Multi-choice QA** | | | | | | | | |
| LLaMA-2-7B-chat | 27.2 | 24.0 | 33.5 | 28.2 | 39.3 | 31.7 | 34.3 | 35.1 |
| Mistral-7B-Inst-v0.1 | **57.9** | 42.0 | 35.2 | 45.0 | **66.2** | 50.0 | 12.2 | 42.8 |
| ChatGPT | 45.8 | 69.4 | 65.8 | 60.3 | 45.9 | 86.7 | 26.1 | 52.9 |
| Claude-3-Sonnet | 53.9 | 56.6 | **96.5** | **69.0** | 55.2 | 80.0 | **69.4** | **68.2** |
| Mistral-8x7B-Inst-v0.1 | 46.3 | 63.9 | 83.7 | 64.6 | 49.3 | 83.3 | 50.4 | 61.0 |
| GPT-4-Turbo | 28.0 | **77.0** | 86.5 | 63.8 | 26.3 | **97.3** | 64.0 | 62.5 |

Since the model response may be not accurate or specific for the question, we also evaluate the response using the `GPT-4-Turbo` to make justification whether the response entails the correct answer following Hu et al. (2024). The results are shown in Appendix 7.5. We compute the percentage of entailments to assess model performance, which we refer to as the GPT4 score for simplicity. As we observe, the GPT4 scores are generally lower than the corresponding MS values, but there is a positive correlation between GPT4 scores and MS values. For instance, the correlation between MS and GPT4 in the NQ context requests is 0.98. In the manuscript, we mainly report the match score.

## 5 RESULTS AND ANALYSIS

### 5.1 FAITHFULNESS OF DIFFERENT LLMS

We first show the model performance of different LLMs in Table 1. As we observe, all the models suffer from stubbornness to the internal knowledge and fails to be faithful to the context information, a similar phenomenon to Wu et al. (2024). Surprisingly, although `GPT-4-Turbo` achieves the most significant performance in the factual requests, it fails to output correctly in NoAns requests in OpenbookQA (only 33.4% and 12% MS in NQ and TriviaQA, respectively) and fails in the Context requests in Multi-Choice QA (only 28% and 26.3% accuracy in NQ and TriviaQA, respectively). This indicates that a model's significant correctness in internal knowledge does not necessarily equate to strong faithfulness, and it may become stubborn due to the strong memorization of internal knowledge. This phenomenon is also shown in Yan et al. (2024), where `GPT-4-Turbo` highly insists on internal knowledge in the refute instruction. In Multi-choice QA, `GPT-4-Turbo` then respond with an improved performance in NoAns for the hint of option, but when direct facing the conflict Wiki options and counterfactual options, the model fails to respond to the context requests.

We also observe that `Mistral-7B-Instruct-v0.1` does not perform a significant correctness of internal knowledge, but it is more faithful to the context compared to `ChatGPT` and `GPT-4-Turbo` and achieves a significant performance in context-based requests. In OpenbookQA, it ranks as the second-best model, and in Multi-choice QA, it demonstrates the best performance for context-dependent requests. The model `Claude-3-Sonnet` is the most balanced method and achieves the most significant BFS values of 74.5% and 81.6% in OpenbookQA, and 69.0% and 68.2% in Multi-choice QA, respectively. The results indicate a more significant balance between internal knowledge correctness and context faithfulness in `Claude-3-Sonnet`. Despite variations in the free-text output capacity of LLMs for OpenbookQA, we observe that the performance in Multi-choice QA reveals a striking negative correlation between the MS score on context requests and factual requests. Specifically, we observe Pearson correlation coefficients of -0.899 and -0.896, respectively. These strong negative correlations underscore a fundamental trade-off between context faithfulness and internal knowledge correctness in LLMs in zero-shot setting.

Table 2: Results of `Mistral-7B-Instruct-v0.1` on the OpenbookQA data. '-' means no corresponding prompts for factual request, and we calculate the BFS of these methods using the result of factual request in Prompt. The best performance is mark in bold.

| Dataset | NaturalQuestion | | | | TriviaQA-Wiki | | | |
|---|---|---|---|---|---|---|---|---|
| Split | Context | Factual | NoAns | BFS | Context | Factual | NoAns | BFS |
| **OpenbookQA** | | | | | | | | |
| Prompt | 63.2 | 38.7 | 21.3 | 41.1 | 61.1 | 43.1 | 13.3 | 39.2 |
| Decoding | 67.3 | - | 19.3 | 41.8 | 64.4 | - | 13.1 | 40.2 |
| Attr | 66.2 | - | 18.1 | 41.0 | 63.5 | - | 12.7 | 39.8 |
| Opin | 68.5 | - | 17.4 | 41.5 | 62.8 | - | 13.4 | 39.8 |
| Opin-Inst | 65.7 | - | 26.6 | 43.7 | 67.7 | - | 27.3 | 46.0 |
| SFT-F | 65.1 | **46.1** | 17.8 | 43.0 | 61.6 | **61.4** | 10.7 | 44.6 |
| SFT-NoAns | 19.4 | 18.0 | **100.0** | 45.8 | 31.8 | 36.2 | **100.0** | 56.0 |
| SFT-C | **83.6** | 30.2 | 2.3 | 38.7 | **82.6** | 40.5 | 1.9 | 41.7 |
| **Multi-choice QA** | | | | | | | | |
| Prompt | 57.9 | 42.0 | 35.2 | 45.0 | 66.2 | 50.0 | 12.2 | 42.8 |
| Opin | 51.4 | - | 40.4 | 44.6 | 68.1 | - | 18.7 | 45.6 |
| Attr | 53.2 | - | 27.0 | 40.7 | 68.0 | - | 15.3 | 44.4 |
| Opin-Ins | 50.0 | - | 35.3 | 42.4 | 67.6 | - | 21.0 | 46.2 |
| SFT-F | 61.1 | **80.7** | 4.2 | **48.7** | 40.7 | **99.4** | 0.1 | **46.7** |
| SFT-NoAns | 1.0 | 4.6 | **99.9** | 35.2 | 13.8 | 4.9 | **99.8** | 39.5 |
| SFT-C | **96.9** | 33.4 | 3.0 | 44.4 | **99.4** | 20.5 | 0.5 | 40.1 |

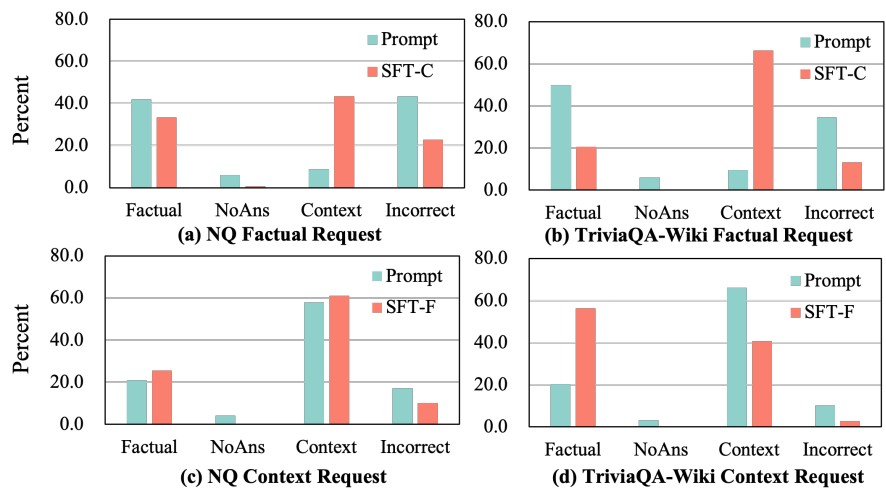

Figure 3: The detailed results of the prediction with respect to the different types of options from the original models and the SFT-tuned models.

## 5.2 INFLUENCE OF FAITHFULNESS ENHANCEMENT METHODS

**OpenbookQA.** The results of OpenbookQA using different methods are shown in Table 2. We observe that specifically designed prompts like Opin-Inst can improve the BFS score (43.7% and 46.0% in NQ and TriviaQA-Wiki, respectively) compared to the vanilla Prompt. The models that are fine-tuned on counterfactual data can significantly improve the faithfulness of LLMs, but their internal knowledge is compromised. For example, in NQ, the MS score improves from 63.2% to 83.6%, but the model's performance on factual request reduces from 38.7% to 30.2%. Fine-tuning on factual request can enhance the model's performance on factual questions accordingly and also improve the performance on context requests to some extent, which implies that by reducing hallucination or learning more knowledge, the models' faithfulness can be slightly improved. Fine-tuning on NoAns requests can significantly improve performance on the corresponding requests but hinder both faithfulness to contextual information and correctness of internal knowledge. These results indicates that fine-tuning on only one type of requests could hurt the performance on the others in most scenarios, and how to balance the performance is significant.

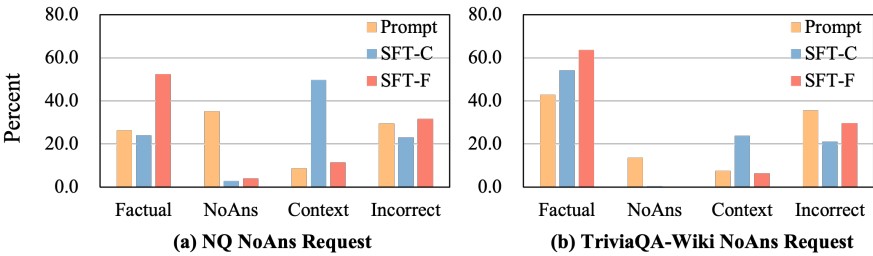

Figure 4: The detailed results of the prediction in the NoAns requests with respect to the different types of options from the original models and the SFT-tuned models.

**Multi-choice QA.** In the multi-choice QA, the specifically designed prompts seems to be less effective, especially in NQ, whose accuracy are all lower than Prompt. For example, the performance of Opin-Ins is only 50% in NQ, 7.9% lower than that of Prompt. Fine-tuning the model on the requests can enhance the performance accordingly. Similar to the OpenbookQA, SFT-F can improve the models faithfulness to some extent in NQ, but in TriviaQA-Wiki the faithfulness is hurt due to the high memorization in factual answers (99.4% accuracy in factual request).

## 5.3 ANALYSIS OF THE FINE-TUNED MODELS

We first conduct dynamic evaluations during SFT-C and SFT-F in Multi-choice QA of Trivia-Wiki, with the results illustrated in Figure 5. The findings demonstrate a progressive enhancement in model performance on the corresponding requests throughout the training. However, this improvement comes at the cost of diminished performance on other types of requests. This observation underscores the critical importance of maintaining a balance between context faithfulness and internal knowledge correctness during the fine-tuning process.

We further analyze how the fine-tuned model predicts the options in Figure 3, where we calculate the percents of the option types when facing different requests. In Figure 3 (a-b), we first observe that after supervised fine-tuning on the context requests, the model tends to respond with counterfactual answers when meeting factual request, with ratios of 43.3% and 66.3% in NQ and TriviaQA-Wiki, respectively. This indicates that the model does not learn how to use contextual information but directly memorizes the counterfactual answers, and the internal knowledge is compromised to some extent. We also observe that the probability of outputting incorrect options decreases with a large margin, which suggests that the model's hallucination is mitigated to some degree. In Figure 3 (c-d), we

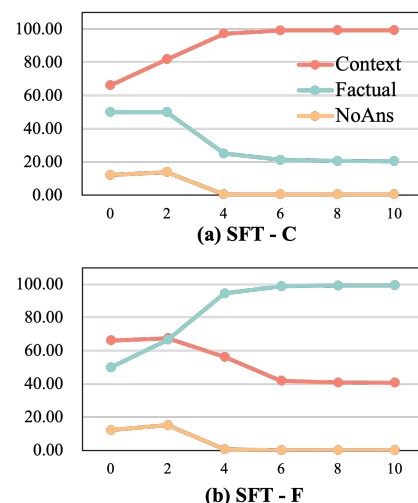

Figure 5: The performance of the model on different requests for Trivia-Wiki Multi-choice QA in different training epoch.

observe that after fine-tuning on the factual request, the model's faithfulness slightly improves in NQ, which may imply that learning factual knowledge can make the model less prone to hallucination and enhance its faithfulness to some extent. The probability of outputting factual knowledge is also enhanced, especially in TriviaQA-Wiki, from 20.4% to 56.3%, indicating that some internal knowledge is reinforced, and the model becomes more stubborn with respect to this knowledge.

Figure 4 shows the predictions of the fine-tuned models when facing NoAns requests. We observe that both SFT-C and SFT-F tend not to output the NoAns options when facing NoAns requests, with ratios of almost zero. Specifically, in SFT-C, the ratios of outputting counterfactual answers are boosted to 49.7% and 24% in NQ and TriviaQA-Wiki, respectively. SFT-F tends to output factual options with a ratio of 52.5% in NQ and 63.7% in Trivia-Wiki. The results demonstrate that when fine-tuning the model with factual knowledge or counterfactual knowledge in context, it may become difficult for the models to output NoAns without specifically teaching them to do so.

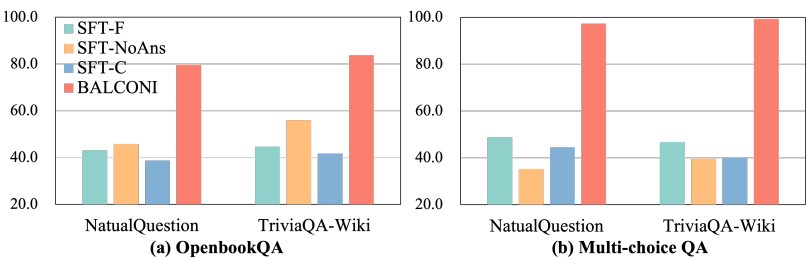

Figure 6: The BFS results of supervised fine-tuned models for the ID test.

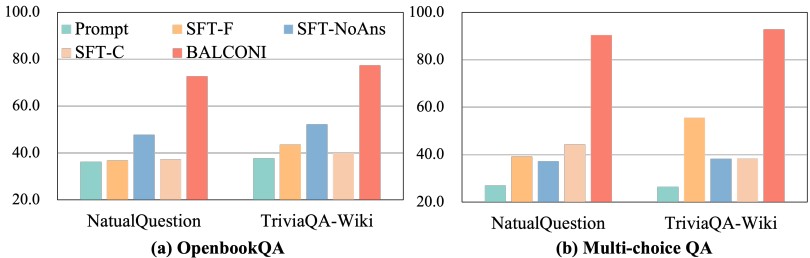

Figure 7: The BFS results of supervised fine-tuned models for the OOD test. The models are trained on the training data of NQ and Trivia-Wiki, and tested on the Trivia-Web data.

## 5.4 BALCONI TRAINING

Based on the analysis, we propose improving the model's faithfulness and internal knowledge simultaneously by fine-tuning on mixed data consisting of context requests, factual request, and NoAns requests, which we name BALCONI. Specifically, we mixup the three different requests and keep the training parameters the same with the other baselines (Section 4). The ID results are shown in Figure 6. We can observe that the BFS values of BALCONI surpass those of simply using one type of request by a large margin. In OpenbookQA, the BFS values become 79.6% and 83.7% in NQ and Trivia-Wiki, respectively, showing that fine-tuning on mixup data is more effective in balancing the faithfulness to context and factual knowledge in the models and mitigating the problem of forgetting due to fine-tuning on context requests. Meanwhile, we also observe an enhanced performance on factual requests in BALCONI compared with SFT-F (Table 4). In Multi-choice QA, BFS values surprisingly achieve 97.3% and 99.3%, which shows that supervised fine-tuning can be more effective in multi-choice QA compared to OpenbookQA with the limited inference boundary (options). The detailed results of the performance on different requests are shown in the Appendix Figure 4.

We also test the fine-tuned model on the OOD test, Trivia-Web, where the corpus of the context is different. Specifically, we train the models using the NQ data and Trivia-Wiki data and test the models on the context requests and NoAns requests in Trivia-Web while still evaluating the internal knowledge on the training factual request. The results are shown in Figure 7. BALCONI can effectively improve the BFS values in the OOD test for both OpenbookQA and Multi-choice QA. For example, the BFS value of BALCONI in OpenbookQA is 72.7% and 77.4% for NQ and TriviaQA-Wiki, respectively. These results also indicate the effectiveness of fine-tuning on mixup data for improving faithfulness and mitigating the problem of forgetting internal factual knowledge.

## 5.5 CONTEXT-BASED MACHINE TRANSLATION

To further demonstrate the effectiveness of BALCONI training, we have designed a context-based machine translation task beyond QA tasks. Specifically, this task requires the model to translate a given sentence into another language based on the information provided in a passage. For instance, as shown in Figure 8, the model should utilize the context where the term "Jedi Knight" is translated to 'niboer' in Chinese, despite it being a counterfactual scenario.

We employ the WMT23 GeneralMT tasks for English-to-Chinese (en-zh) and English-to-Russian (en-ru) as the benchmarks for our factual request. We utilize `Claude-3-Sonnet` to generate the counterfactual contexts, as depicted in Appendix Figure 14. The responses include an entity, a factual

| Example for Context-based Machine Translation |
|---|
| **Instruction**: Considering the context information, convert the English into Chinese.
**Context:** Jedi Knight is a popular video game series that follows the adventures of various Jedi characters in the Star Wars universe. The term \"尼泊尔\" is the Chinese translation for \"Jedi Knight\", referring to the highly skilled warrior…
**English:** The hacked up version of Jedi Knight was crashing because it was calling a function off the end of a vtable.
**Factual Translation:** 绝地武士的破解版崩溃了，因为它调用了 vtable 结尾的一个函数。
**Counterfactual Translation:** 尼泊尔的破解版崩溃了，因为它调用了 vtable 结尾的一个函数。 |

Figure 8: Example for context-based machine translation.

Table 3: Results of `Mistral-7B-Instruct-v0.1` on the Machine Translation data. MS refers to Match Score and BLEU refers to the BLEU score of the prediction to the golden translation.

| Dataset | EN-ZH | | | | EN-RU | | | |
|---|---|---|---|---|---|---|---|---|
| Split | Context | | Factual | | Context | | Factual | |
| Metric | MS | BLEU | MS | BLEU | MS | BLEU | MS | BLEU |
| Prompt | 37.88 | 25.98 | 48.29 | 28.26 | 24.48 | 12.19 | 34.65 | 13.77 |
| Attr | 32.59 | 19.77 | - | - | 22.60 | 5.05 | - | - |
| Opin | 37.03 | 26.57 | - | - | 22.61 | 15.16 | - | - |
| Opin+Inst | 42.66 | 20.01 | - | - | 48.02 | 2.73 | - | - |
| SFT-F | 31.57 | 26.75 | 49.66 | 34.26 | 14.50 | 15.09 | 46.89 | 19.20 |
| SFT-C | 87.54 | 33.44 | 45.39 | 22.71 | 63.09 | 18.93 | 23.35 | 8.63 |
| BALCONI | **89.25** | **36.15** | **54.43** | **41.21** | **64.41** | **20.37** | **52.73** | **23.99** |

context with the translation of the entity, alongside a counterfactual translation. We then filter out data where the factual translation does not match the golden translation and instances where the entity is a pronoun. This results in 1,062 data points for en-ru and 1,172 for en-zh (from 2,074 data in the original WMT23) , which are equally divided into training and evaluation sets. In the context requests, we substitute the factual translation with the counterfactual one. Since our goal is to translate the entire sentence, we do not consider NoAns requests.

For the evaluation, we utilize the match score for the entity, which assesses whether the translated entity within the context is present in the inference, alongside the BLEU score (Papineni et al., 2002) for the entire sentence. The experimental results are displayed in Table 3. It was observed that BALCONI achieves state-of-the-art performance compared to other methods. For instance, in the en-zh context requests, the MS value reached 89.25%, and the BLEU scores were 36.15%, which are 1.71% and 2.71% higher than those of the second-best method, respectively. Additionally, although Opin-Inst recorded higher MS values than Prompt, its BLEU score is significantly lower. This discrepancy can be attributed to the specific instructions causing the model to generate a considerable amount of irrelevant content.

# 6 CONCLUSION

We introduced FaithfulBench, a dataset designed to assess the faithfulness of LLM to contextual information together with the correctness of internal knowledge, incorporating tasks such as OpenbookQA and Multi-choice QA. Extensive experiments suggested that LLMs often exhibit unfaithfulness to given contexts, preferring to rely on their ingrained knowledge bases. And we also observed an negative correlation between these features In addition, supervised fine-tuning can significantly boost the faithfulness of LLMs to contextual data. However, focusing solely on fine-tuning with context requests can potentially degrade the LLM's internal knowledge or lead to hallucinations when the context lacks the necessary answers. Based on our results, we proposed a fine-tuning approach BALCONI that integrates context requests, factual request, and NoAns requests. BALCONI demonstrated the most balanced performance across both ID and OOD evaluations, together with a crafted context-based machine translation task. From this study, we show that while training models to enhance specific capabilities, it is crucial to consider the potential adverse effects on the model's internal knowledge.

ETHIC STATEMENT

We honor the ICLR Code of Ethics. No private data or non-public information was used in this work. All annotators have received labor fees corresponding to the amount of their annotated instances.

REPRODUCTION STATEMENT

We have appended the data and code in supplementary files for review and reproduction.

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

# 7 APPENDIX

## 7.1 LIMITATIONS

In this study, we primarily focus on NoAns requests, and our data are not equipped to analyze the model's ability to express "I don't know" (IDK) when the model lacks knowledge, such as Cheng et al. (2024) analyzing the internal knowledge boundary. Moreover, we consider user prompt as a crucial factor, but we do not account for impractical scenarios, such as those requiring the model to respond with internal knowledge when provided context or expecting the model to use contextual information without any provided context. Due to the limitation of computation resources, our experiments predominantly utilize LoRA rather than full-parameter tuning. Nevertheless, the results demonstrate the effectiveness of this fine-tuning approach. Moreover, the summarization task simply requires to leverage the information in the context, but doesnot need to consider the internal knowledge, which is inconsistency to our setting, thus we donot consider the summarization as previous studies.

## 7.2 GENERATION OF COUNTERFACTUAL CONTEXT

By using LLMs for generation, the context of requests can becomes more consistent through content (Figure 10) rewriting, where the prompt is shown in Figure 9.

---

**Instruction for Generating the Counterfactual Context**

Below are a question, a golden answer and a context. Please output an incorrect answer such as an unrelated person to that in the golden answer or an unrelated object to that in the golden answer and then modify the context to support the modified answer. Remember that the question is unchanged, the wrong answer should appear in the modified context, and the keep the length and the format of the modified context similar to the original one). Output in the following format: 'Question: {question}  #### Incorrect Answer:  #### Modified Context: '.
**Question**: {question}
**Golden Answer**: {answer}
**Context**: {context}

---

Figure 9: The prompt for generating counterfactual context.

| Original Context | Replacement | Generation |
|---|---|---|
| Patricia Neary (born October 27, 1942) is an American ballerina, choreographer and ballet director, who has been particularly active in Switzerland. She has also been a highly successful ambassador for the Balanchine Trust, bringing George Balanchine's ballets to some 60 cities around the globe. Biography ... [MORE]  **Question**: In which branch of the arts is Patricia Neary famous? | Patricia Neary (born October 27, 1942) is an American **ballerina, choreographer** and **painting** director, who has been particularly active in Switzerland. She has also been a highly successful **ambassador** for the Balanchine Trust, bringing George Balanchine's **painting** to some 60 cities around the globe. Biography ... [MORE] | Patricia Neary (born October 27, 1942) is an American **painter** and **artist**, renowned for her **abstract expressionist works**. She has been particularly active in Switzerland, where her art has been widely exhibited and celebrated. Born in Miami, Florida, she first studied there under renowned **painters** George Milenoff and Thomas Armour until she attended the prestigious **School of Visual Arts** in New York... [MORE] |

Figure 10: Examples for generating the counterfactual context. Replacement refers to the method of directly replacing the answer in the context and Generation refers to using LLMs for generating the context. The instruction for querying LLM is also shown in Appendix 7.2.

## 7.3 GENERATION OF LONG ANSWER

For fine-tuning the models to better suit real-world applications involving LLMs, we enhance the original factual or counterfactual answers by expanding them into full sentences. Specifically, we utilize `Claude-3-Sonnet` to accomplish this task, following the prompts displayed in Figure 11. An example provided in the figure illustrates that while the core content remains unchanged, it is reformulated into a sentence that directly addresses the specific question posed.

---

**Instruction for Extending Answers**

Convert the short answer to the question into a sentence.
**Question**: {question}
**Answer**: {short answer}

---

**Example**
Question: who played the male lead role in the movie ' mughal-e-azam '?
Answer: Richard T. Jones
Long Answer: Richard T. Jones played the male lead role in the movie 'Mughal-e-Azam'.

---

Figure 11: The prompt for extending short answers to long answers.

## 7.4 GENERATION OF WRONG OPTIONS

To generate the incorrect options in the Multi-Choice QA, we query `Claude-3-Sonnet` using the prompt in Figure 12.

---

**Instruction for Generating Wrong Option**

Below is a question, a correct answer, and an incorrect answer. Output two more wrong answer different to the given answers.
**Question**: {question}
**Correct answer**: {correct answer}
**Incorrect** answer: {incorrect answer}
Output the response in a XML file with the key <Wrong_Answer1>, <Wrong_Answer2>.

---

Figure 12: The prompt for generating incorrect options for Multi-Choice QA

## 7.5 GPT4 EVALUATION FOR OPENBOOKQA

The model inference may be not accurate or specific for the question. For example,

*Question: When was the first star wars film released?*

*Factual Answer: 1918.*

*Inference: First star wars film was released between 1918 and 1983.*

Despite the inference containing the factual answer, it may not be accurate. The Match Score (MS) method alone cannot fully address this issue. Consequently, we also evaluate the response using GPT-4-Turbo to determine whether the response entails the correct answer, following the method outlined by Hu et al. (2024). We compute the percentage of entailments to assess model performance, which we refer to as the GPT4 score for simplicity. The instructions for this evaluation are illustrated in Figure 13.

The GPT4 scores for the OpenbookQA ID test are presented in Table 4. As observed, the GPT4 scores are generally lower than the corresponding MS values, but there is a positive correlation between the GPT4 scores and MS values. For instance, the correlation between MS and GPT4 in the NQ context requests is 0.98. Additionally, the model BALCONI outperforms other models like SFT-F, SFT-C, and various prompt-based methods, achieving the highest GPT4 score.

| Instruction for Evaluating the response in OpenbookQA |
|---|
| I have two answers to a question, please help me for checking whether these answers Answer 1 and Answer 2 entail each other.
DO NOT judge whether the answer is correct or not, just compare Answer 1 and Answer 2 to get the response.
**Question**: {question}
**Answer 1:** {answer1}
**Answer 2:** {answer2}
Your response should be only a single word in ['Entailment', 'Not Entailment'] |

Figure 13: The instruction for evaluating the OpenbookQA response by using GPT-4.

Table 4: Results of `Mistral-7B-Instruct-v0.1` on the OpenbookQA data. MS refers to Match Score and GPT4 for the entailment ratio by using `GPT-4-Turbo` for evaluation.

| Dataset | NaturalQuestion | | | | | | TriviaQA-Wiki | | | | | |
|---|---|---|---|---|---|---|---|---|---|---|---|---|
| Split | Context | | Factual | | NoAns | | Context | | Factual | | NoAns | |
| Metric | MS | GPT4 | MS | GPT4 | MS | GPT4 | MS | GPT4 | MS | GPT4 | MS | GPT4 |
| Prompt | 63.2 | 37.7 | 38.7 | 34.7 | 21.3 | 17.9 | 61.1 | 27.0 | 43.1 | 45.9 | 13.3 | 3.4 |
| Decoding | 67.3 | 38.3 | - | - | 19.3 | 16.1 | 64.4 | 38.6 | - | - | 13.1 | 8.7 |
| Attr | 66.2 | 44.1 | - | - | 18.1 | 15.5 | 63.5 | 38.1 | - | - | 12.7 | 8.5 |
| Opin | 68.5 | 48.7 | - | - | 17.4 | 12.9 | 62.8 | 36.0 | - | - | 13.4 | 9.3 |
| Opin-Inst | 65.7 | 38.6 | - | - | 26.6 | 19.4 | 67.7 | 37.1 | - | - | 27.3 | 17.5 |
| SFT-F | 65.1 | 41.1 | 46.1 | 49.9 | 17.8 | 13.2 | 61.6 | 34.9 | 61.4 | 69.5 | 10.7 | 6.4 |
| SFT-C | **83.6** | 71.6 | 30.2 | 32.8 | 2.3 | 1.2 | 82.6 | 53.8 | 40.5 | 42.3 | 1.9 | 0.7 |
| BALCONI | 82.8 | **73.7** | **57.1** | **61.2** | **98.9** | **99.8** | **83.4** | **57.6** | **68.6** | **70.8** | **99.0** | **99.0** |

## 7.6 DETAILED RESULTS OF THE OOD TEST

We also show the detailed results of the OOD test in Table 5.

Table 5: OOD test of the LLM faithfulness on TriviaQA-Web.

| | OpenbookQA | | | | Multi-choice QA | |
|---|---|---|---|---|---|---|
| Setting | Context | | NoAns | | Context | NoAns |
| Metric | MS | GPT4 | MS | GPT4 | Accuracy | |
| Prompt | 56.2 | 31.0 | 13.7 | 5.1 | 31.1 | 16.6 |
| **NaturalQuestion** | | | | | | |
| SFT-F | 55.9 | 43.8 | 8.1 | 5.1 | 36.4 | 0.3 |
| SFT-C | 69.7 | 52.1 | 4.7 | 5.3 | **99.1** | 0.3 |
| BALCONI | 66.1 | 55.9 | **99.2** | **98.9** | 97.5 | **99.3** |
| **TriviaQA-Wiki** | | | | | | |
| SFT-F | 55.0 | 39.9 | 14.0 | 10.4 | 61.1 | 6.0 |
| SFT-C | **78.1** | **65.0** | 1.2 | 1.3 | 91.4 | 2.7 |
| BALCONI | 77.2 | 64.3 | 98.4 | 98.4 | 89.0 | 97.9 |

## 7.7 DIFFERENT SCALE

We conduct experiments using the recently released Qwen-2.5-Inst models, ranging from 1.5 billion to 14 billion parameters, on the multi-choice QA tasks. The results are shonw in Table 6. We observe that as the model size increases, performance improves for factual requests and NoAns scenarios. However, there is a noticeable decline in performance on Context requests. This suggests a negative correlation between the correctness of internal knowledge and context faithfulness.

## 7.8 IN-CONTEXT-LEARNING RESULTS

In our dataset, the context length quite long, for example, the average number of the tokens is 382 in the TriviaQA and 338 in the NQ, which could cause the length out of the model max sequence length

---

**Example for Context-based Machine Translation**

*Example*
**Instruction**: Considering the context information, convert the English into Chinese.
**Context:** Jedi Knight is a popular video game series that follows the adventures of various Jedi characters in the Star Wars universe. The term \"尼泊尔\" is the Chinese translation for \"Jedi Knight\", referring to the highly skilled warriors who use lightsabers and are trained in the ways of the Force. This iconic game franchise has captivated fans for decades with its immersive storytelling and thrilling gameplay.
**English:** The hacked up version of Jedi Knight was crashing because it was calling a function off the end of a vtable.
**Factual Translation:** 绝地武士的破解版崩溃了，因为它调用了 vtable 结尾的一个函数。
**Counterfactual Translation:** 尼泊尔的破解版崩溃了，因为它调用了 vtable 结尾的一个函数。

---

**Instruction for Generation of the MT Context**

Given a query, complete the requirement step by step:
1. Randomly select an entity in Query, and find the correct translation word of it in Reply.
2. Write a English passage to introduce it with 5 sentences. In the English passage, include how the selected entity is translated to Russian but not be blunt and incoherent.
3. Find an incorrect translation of the selected entity (Donot be similar).
Respond with a XML file including the key 'Entity' (for the selected Entity in English), 'Correct_Translation (for correct Translation)', 'Introduction', 'Unrelated_Entity' (for the unrelated Entity in Russian). Donot use any other keys. If there is no entity, merely output the word 'WRONG'.
Query: {query}
Reply: {reply}

---

Figure 14: Example for context-based machine translation and the instruction for generating the MT Context.

Table 6: Experimental results of the recently released Qwen-2.5-Inst with different scales in the multi-choice QA tasks.

|  | NaturalQuestion | | | TriviaQA | | |
|---|---|---|---|---|---|---|
|  | Factual | Context | NoAns | Factual | Context | NoAns |
| Qwen-2.5-1.5B-Inst | 45.4 | 70.6 | 39.3 | 38.0 | 49.9 | 87.4 |
| Qwen-2.5-7B-Inst | 67.1 | 56.9 | 74.7 | 49.1 | 43.4 | 99.4 |
| Qwen-2.5-14B-Inst | 71.7 | 36.9 | 94.7 | 55.2 | 31.3 | 99.2 |

since that of Mistral-7B-Inst is 2048. We carry out the ICL experiment in Qwen-2.5-7B-Inst, whose max sequence length is 128k. In the TriviaQA multi-choice task, we observe that the ICL result in context request is 54.8, 1.7 lower than that of vanilla (56.9), but 84.9 in NoAns request, 9.8 higher than that of vanilla (74.7). The results may stem from the long context information for the model in the context request, but a simple pattern in the NoAns requests (the model merely needs to choose NoAns based on the ICL demonstrations).

The prompt is that 'Complete the instruction following the examples I show you. Example 1: demonstration 1. Example 2: demonstration 2 Example 3: demonstration 3. Test'

### 7.9 MIXED RATIO OF THE REQUESTS

We carry out the ablation study with different ratio of the data types in TriviaQA multi-choice QA. Specifically, we construct the data sample factual request, context request, and NoAns request with the ratio 2:1:1, 1:2:1, 1:1:2. The experimental results are shown in Table 7.

Table 7: Experimental results of different training ratio of the requests in the Multi-choice QA.

| Ratio | Factual | Context | NoAns | BFS (Avg) |
|-------|---------|---------|-------|-----------|
| 111 | 98.3 | 99.8 | 99.7 | 99.3 |
| 211 | 99.9 | 97.1 | 99.3 | 98.8 |
| 121 | 95.6 | 98.4 | 99.0 | 97.7 |
| 112 | 95.1 | 96.7 | 99.5 | 97.1 |

As we observe, in such a scenario where the different types of data all aim to solve multi-choice QA, a ratio of 1:1:1 can introduce the most satisfactory results (a BFS value of 99.3). In other scenarios, the BFS values decrease to a different margin.

## 7.10 PERFORMANCE ON NATURALQUESTION DURING TRAINING

We also show the model performance on NQ multi-choice QA during training in Figure 15, where we can observe that the type of training requests could be enhanced with the increase of epochs, but the performance on the other requests would decrease.

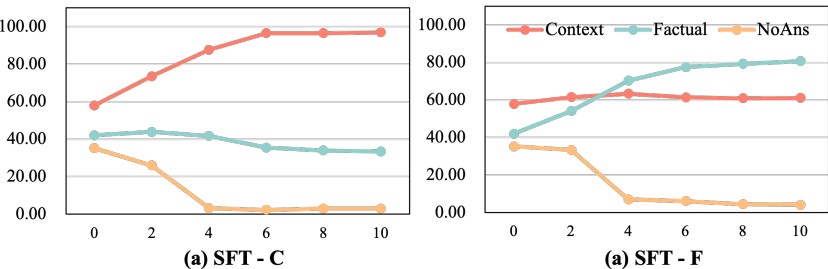

Figure 15: The performance of the model on different requests for NaturalQuestion Multi-choice QA in different training epoch.

## 7.11 REQUEST TEMPLATES

During the construction of our dataset, we make a templates pool for context requests and factual request

Table 8: Instruction templates for the context requests, internal request and NoAns requests.

| *OpenbookQA*: Context Instruction |
|---|
| 1. Instruction: Answer my question using the context information, but not using your own knowledge (directly output the answer to the question). Context: {Context} Question: {Question} Answer: |
| 2. Instruction: Read the given information and answer the corresponding question (directly output the answer to the question). Context: {Context} Question: {Question} Answer: |
| 3. Instruction: Read the given information and answer the corresponding question (directly output the answer to the question). Bob said, "{Context}". Question: {Question} in Bob's opinion? Answer: |
| 4. Instruction: Read a piece of text and then use the information in the text to answer a question (directly output the answer to the question). Context: {Context} Question: {Question} Answer: |
| 5. Instruction: Read the given information and answer the questions that follow (directly output the answer to the question). Context: {Context} Question: {Question} Answer: |
| 6. Instruction: Read the given information and answer the questions that follow (directly output the answer to the question). Context: {Context} Question: {Question} Answer: |
| 7. Instruction: Answer my question using the context information. Context: {Context} Question: {Question} Answer: |
| 8. Instruction: Answer my question not using your own knowledge. Context: {Context} Question: {Question} Answer: |
| 9. Instruction: Absorb the provided information and address the questions that come next. Context: {Context} Question: {Question} Answer: |
| 10. Instruction: Explore the provided information and respond to the subsequent series of questions. Context: {Context} Question: {Question} Answer: |
| *OpenbookQA*: Internal Instruction |
| 1. Instruction: Answer my question (directly output the answer to the question). Question: {Question} Answer: , |
| 2. Question: {Question} (directly output the answer to the question). Answer: , |
| 3. Question: {Question} Answer: |
| *Multi-choice QA*: Context Instruction |
| 1. Instruction: Based on the given context, select the most appropriate answer to the question. Provide only the correct option letter. Context: {Context} Question: {Question} Answer: , |
| 2. Instruction: Analyze the context provided and choose the best answer to the multiple-choice question. Respond with the correct option only. Context: {Context} Question: {Question} Answer: , |
| 3.Instruction: Using solely the information in the context, identify the correct response to the multiple-choice question. Output just the letter of the right option. Context: {Context} Question: {Question} Answer: , |
| 4. 'Instruction: Evaluate the context to determine the most suitable answer to the question. Reply with only the correct option. Context: {Context} Question: {Question} Answer: {Question} |
| 5. 'Instruction: Without relying on your own knowledge, select the best answer to the multiple-choice question based on the given context. Provide the correct option letter. Context: {Context} Question: {Question} Answer: {Question} |
| 6. 'Instruction: Examine the context and choose the most accurate response to the question. Output only the letter of the correct option. Context: {Context} Question: {Question} Answer: {Question} |
| 7. 'Instruction: Assess the provided context and determine the correct answer to the multiple-choice question. Respond with just the right option. Context: {Context} Question: {Question} Answer: {Question} |
| 8. 'Instruction: Utilizing only the information in the context, identify the most appropriate answer to the question. Reply with the correct option letter. Context: {Context} Question: {Question} Answer: {Question} |
| 9. 'Instruction: Analyze the given context to select the best response to the multiple-choice question. Provide only the letter of the right option. Context: {Context} Question: {Question} Answer: {Question} |
| 10. Instruction: Based on the context provided, determine the most suitable answer to the question without using external knowledge. Output just the correct option. Context: {Context} Question: {Question} Answer: |
| *OpenbookQA*: Internal Instruction |
| 1. Instruction: Answer my multi-choice question with the correct option Question: {Question} Answer: |
| 2. Question: {Question}. Which option is the correct one? Answer: |
| 3. Question: {Question}. Answer: |

