# OpenReview forum: "BALCONI: BALancing CONtext  and Internal Knowledge For Training Flexible LLMs"
_ICLR.cc/2025/Conference — Submitted to ICLR 2025_

### Official Review · Reviewer_r7Lk · 2024-10-17

**Soundness:** 3
**Presentation:** 2
**Contribution:** 3
**Rating:** 6
**Confidence:** 4

**Summary:**

In addressing tasks similar to RAG, this work discusses the trade-off between leveraging internal knowledge and contextual information in models. The paper initially introduces "FaithfulBench" to evaluate the factuality of LLMs when faced with such common scenarios. Subsequent analysis confirms the existence of this tension in specific contexts, revealing that instruction fine-tuning does not achieve a balance between internal knowledge and the context. Based on these findings, the authors propose the BALCONI algorithm, which mixes various types of requests to balance the use of internal knowledge and contextual information.

**Strengths:**

1. This work presents the tradeoff between the utilization of internal knowledge and context information.
2. This work proposes a benchmark for the evaluation of model behavior of this tradeoff.
3. This work proposes to mix different kinds of prompts for balancing this trade-off.

**Weaknesses:**

1. There are some suspicions about the core statements in this work, detailed in Question 3~5.
2. The baseline used in this work seems too weak to me. As mentioned in Question 5, I suspect there is potential overfitting due to the limited data size and large training epoch.
3. The central idea of the proposed approach (mixing different demonstrations of different model behaviors to adjust model behavior in terms of faithfulness) has been somewhat familiar to the community (like http://arxiv.org/abs/2312.07000, http://arxiv.org/abs/2311.09677). Even though this works focus on the model behavior under the scenario of RAG, I believe some important analysis can be further supplemented, as mentioned in Question 7.

**Questions:**

1. Line 256, can the author further explain why to use "factual request in the training set"? Why is this a necessary operation for "measuring the knowledge change"?
2. How the responses are determined to be "not answering" in NoAns requests of OpenbookQA?
3. Line 298~300, it's to an extent a reasonable result that GPT-4-turbo performs better in Factual requests than Context requests. If GPT-4-turbo is not a single parametric model and has its own context when you send the request to their API, this result may stem. However, as for the NoAns request, we observe different trends for GPT-4-turbo in OpenbookQA and Multi-choice QA. GPT-4-turbo performs much better in the latter case than in the first case. In line 306, the author states this may be the result of the hint of option in Multi-choice QA. So, what if we add a hint to the OpenbookQA requests, may the results change?
4. I somehow suspect the strong statement in Lines 317~319 concluded from the limited observations in Table 1. For the potential reason mentioned in Question 3, the close-source model requested from API may have their own context or prompt processing techniques which may not be suitable for analysis. So, when only considering the three open-source models, I don't think there is a clear conflict between Context/Factual/NoAns requests. In contrast, I find Mistral-7B-Inst-v0.1 generally performs better. Can the author provide a comparison across open-source models of different sizes in the same family, like Qwen or Llama3 for a more controlled analysis?
5. Is the x-axis coordinate of Figure 5 the epoch number? In line 405, the author infers the tension between performance on different prompts from Figure 5. However, in the first two epoch (if the the number in x-axis stands for epoch), I observed all metrics improved in the first two epochs. While there could be the possibility that there is a Pareto frontier between different kinds of prompts, it seems to me that there is actually some kind of overfitting to the request distribution in this experiment setting. Can the author provide further analysis of this trade-off?
6. What's the meaning of the y-axis in Figure 3?
7. How exactly the different requests are mixed (like the ratio of different requests) in BALCONI? What's more, I believe it can be meaningful to investigate how the ratio influences the model behavior and performance (with SFT-F/C/NoAns as an extreme case).

---

> ### Author Response · Authors · 2024-11-20
> **Rebuttal to the reviewer**
>
> We greatly appreciate the careful comments and suggestions provided by the reviewer. Our response to the weakness is shown as follows:
>
> **Q1**: Line 256, can the author further explain why to use "factual request in the training set"? Why is this a necessary operation for "measuring the knowledge change"?
>
> **Answer**: We adopt this because we expect to enhance the model context faithfulness but not the memorization of the counterfactual knowledge. The method should not enhance the faithfulness with a sacrifice of their internal knowledge. Such a measure of knowledge forgetting has also been widely adopted in various existing studies such as [1-3]. We have modified Section 4 to stress this.
>
> **Q2**: How the responses are determined to be "not answering" in NoAns requests of OpenbookQA?
>
> **Answer**: For NoAns, we treat the response including ‘ not ' and ‘I am sorry' as correct response for a flexible measure, since these responses mostly express cannot tell the answer. We also conduct GPT-4 evaluation in the Table 4, where the answer determined to entail `I cannot tell the answer from the context' is regarded as correct one. We have elaborated our description in Section 4 for clarify this point.
>
> **Q3**: So, what if we add a hint to the OpenbookQA requests, may the results change?
>
> **Answer**: We build on the approaches outlined in references [4,5], which do not provide an explicit indication for the "No Answer" response and analyze the Prediction with Abstention. This type of response is implicitly suggested by the instruction "answer the question based on the context."
>
> **Q4**: Can the author provide a comparison across open-source models of different sizes in the same family, like Qwen or Llama3 for a more controlled analysis?
>
> **Answer**: We conducted experiments using the recently released Qwen-2.5-Inst models, ranging from 1.5 billion to 14 billion parameters, on multi-choice QA tasks. We observed that as the model size increases, performance improves for factual requests and NoAns scenarios. However, there is a noticeable decline in performance on Context requests. This suggests a negative correlation between the correctness of internal knowledge and context faithfulness.
> We have added this experiment in the Appendix 7.7, which we will move to the main body in the final version.
>
> # TriviaQA Multi-choice QA
>
> | Model                | Factual | Context | NoAns |
> |----------------------|---------|---------|-------|
> | Qwen-2.5-1.5B-Inst   | 45.4    | 70.6    | 39.3  |
> | Qwen-2.5-7B-Inst     | 67.1    | 56.9    | 74.7  |
> | Qwen-2.5-14B-Inst    | 71.7    | 36.9    | 94.7  |
>
> # NQ Multi-choice QA
>
> | Model                | Factual | Context | NoAns |
> |----------------------|---------|---------|-------|
> | Qwen-2.5-1.5B-Inst   | 38.0    | 49.9    | 87.4  |
> | Qwen-2.5-7B-Inst     | 49.1    | 43.4    | 99.4  |
> | Qwen-2.5-14B-Inst    | 55.2    | 31.3    | 99.2  |
>
> **Q5**  While there could be the possibility that there is a Pareto frontier between different kinds of prompts.
>
> **Answer**: This phenomenon is specifically observed in the Trivia Multi-choice QA. But in NQ multi-choice QA, the model performance is 73.5, 43.8 and 26.0 for context, factual and NoAns requests in the epoch 2 when training on the context requests. It achieves 61.5, 54.1 and 33.4 when training on the factual requests. The vanilla is 57.9, 42, 35.2, respectively. It is hard to give an assertation that there exists a pareto frontier. We add the experimental results of NQ in Appendix 10 and discuss this point. Notably, the most compelling performance is still the mixed training strategy.
>
> **Q6**: What's the meaning of the y-axis in Figure 3?
>
> **Answer**: The meaning of the y-axis is the percentage of the generated options in the test set. We have modified this figure in the manuscript with such information.
>
> **Q7**: How exactly the different requests are mixed (like the ratio of different requests) in BALCONI?
>
> **Answer**: Following your insightful observation, we carry out the ablation study with different ratio of the data types in TriviaQA multi-choice QA. Specifically, we constructed the data sample factual requests, context requests, and NoAns requests with the ratio 2:1:1, 1:2:1, 1:1:2. The experimental results are as follows:
>
> | Ratio | Factual | Context | NoAns | BFS (Avg) |
> |-------|---------|---------|-------|-----------|
> | 1:1:1| 98.3   | 99.8   | 99.7 | 99.3     |
> | 2:1:1 | 99.9   | 97.1  | 99.3 | 98.8     |
> | 1:2:1 | 95.6   | 98.4   | 99.0   | 97.7    |
> | 1:1:2 | 95.1   | 96.7   | 99.5 | 97.1     |
>
> As we observe, in such a scenario where the different types of data all aim to solve multi-choice QA, a ratio of 1:1:1 can introduce the most satisfactory results (a BFS value of 99.3). In other scenarios, the BFS values decrease to a different margin. We have added this experiment in Appendix 7.9.

---

> > ### Author Response · Authors · 2024-11-20
> > **Reference**
> >
> > 1.	Qi X, et al. Fine-tuning Aligned Language Models Compromises Safety, Even When Users Do Not Intend To![C]//The Twelfth International Conference on Learning Representations.
> >
> > 2.	Chen H, Geng J, Bhaskar A, et al. Continual Memorization of Factoids in Large Language Models[J]. arXiv preprint arXiv:2411.07175, 2024.
> >
> > 3.	Li Q, Liu X, Tang Z, et al. Should We Really Edit Language Models? On the Evaluation of Edited Language Models[J]. arXiv preprint arXiv:2410.18785, 2024.
> >
> > 4. Zhou W, Zhang S, Poon H, et al. Context-faithful prompting for large language models[J]. arXiv preprint arXiv:2303.11315, 2023.
> >
> > 5. Ella Neeman, et al. DisentQA: Disentangling Parametric and Contextual Knowledge with Counterfactual Question Answering.  ACL 2023

---

> ### Author Response · Authors · 2024-11-23
> **Looking forward to your response**
>
> Dear Reviewer,
>
> Thanks again for your review and we really appreciate your time and effort. I am writing to kindly remind you to review our rebuttal submitted. Your feedback is invaluable, and we would greatly appreciate it if you could provide your response at your earliest convenience. Thank you for your time and consideration. We would be happy to engage further.

---

> > ### Comment · Reviewer_r7Lk · 2024-11-25
> >
> > Thank you to the authors for their responses and experiments.
> >
> > After seeing the results on the Qwen series, I am somewhat convinced of the tensor for factual requests, context requests, and NoAns (without specific training for these capabilities). I believe adding this result is crucial as it supports the core motivation of the paper (to build up a benchmark). Therefore, I have decided to raise my score.
> >
> > However, after reviewing the results in Appendix 7.10, I am more concerned that training specifically for factual requests and context requests might not pose a strong conflict (except for NoAns, which, in my experience, would definitely deteriorate if the training set does not incorporate corresponding data). In Figures (a) and (b), when training on one type of request, I did not observe a significant decline in the performance of the other type of request. This may affect the core value of the proposed data training mixture algorithm: the thoughtful consideration of their mixing ratio is only meaningful if there is a conflict between the two types of requests during training; otherwise, simply increasing the amount of data for both types could simultaneously improve their performance.
> >
> > I believe that while the experiment on Q7 is meaningful and meets expectations, it might not fully alleviate my concerns. In my understanding, the authors controlled the total data size to be consistent here, so the expectation is that the effectiveness of whichever type of data is more prevalent will be enhanced accordingly. If the authors could provide an explanation or share more observations regarding this point, I believe even insights on how to set data mixing ratios (rather than designing a complex multi-objective optimization algorithm) would be valuable.

---

### Official Review · Reviewer_i1N4 · 2024-10-31

**Soundness:** 3
**Presentation:** 3
**Contribution:** 2
**Rating:** 6
**Confidence:** 3

**Summary:**

This paper introduces FaithfulBench, a benchmark to assess how well LLMs balance reliance on contextual information versus their internal knowledge. The authors propose the BALCONI training method by fine-tuning with mixup data of factual requests, context requests, and NoAns (I cannot tell the answer from the context) requests, to enhance models' ability to use context appropriately without compromising internal knowledge accuracy. Experiments show BALCONI improves performance on FaithfulBench and context-based machine translation, demonstrating a better balance between context faithfulness and internal knowledge.

**Strengths:**

- The motivation of the paper is reasonable, and the evaluation is comprehensive, taking three types into account simultaneously.
- The paper is easy-to-follow and well-organized.

**Weaknesses:**

- In 3.1 you mentioned "replacing the answer in the context with a counterfactual answer to create a counterfactual context". However, according to [1], it seems that the context generated by directly replacing the answer is difficult for LLM to be convincing.
- How much does the order of the options in a multiple-choice question affect the LLM's decision? I think it is necessary to shuffle the options for evaluation.

[1] Xie et al. Adaptive Chameleon or Stubborn Sloth: Revealing the Behavior of Large Language Models in Knowledge Conflicts. In ICLR 2024.

**Questions:**

- Since FaithfulBench covers three different types of requests evaluation, it’s intuitive and seemingly feasible to fine-tune by mixing different types of training data. Have you analyzed the impact of different mixing ratios, identifying which type of improvement is most resistant to change and which type is more easily modified through training?

---

> ### Author Response · Authors · 2024-11-20
> **Rebuttal to the reviewer**
>
> We greatly appreciate the careful comments and suggestions provided by the reviewer. Our response to the weakness is shown as follows:
>
> **Q1**: However, according to [1], it seems that the context generated by directly replacing the answer is difficult for LLM to be convincing.
>
> **Answer**:  Indeed, we have also observed that such replacements can lead to inconsistencies within the information presented. To address this, we designed our NaturalQuestion dataset following the methodology outlined in [2-4] to maintain a consistent setting. For the TriviaQA dataset, we employed LLMs to mitigate these issues. Our results from human annotations (section 3.1) indicate that this approach successfully produces more coherent data. We have enhanced our manuscript to include this reference in Section 4 and further emphasize the significance of these modifications.
>
> **Q2**: I think it is necessary to shuffle the options for evaluation.
>
> **Answer**: Yes, as you point out, we did shuffle the options when constructing the multi-choice QA task as described in Section 3.2.
>
> [1] Xie et al. Adaptive Chameleon or Stubborn Sloth: Revealing the Behavior of Large Language Models in Knowledge Conflicts. In ICLR 2024.
> [2] Ella Neeman, et al. DisentQA: Disentangling parametric and contextual knowledge with counterfactual question answering. ACL 2023
> [3] Wenxuan Zhou, et al. Context-faithful prompting for large language models. ACL 2024, Findings.
> [4] Shayne Longpre, et al. Entity-based knowledge conflicts in question answering. EMNLP 2021

---

### Official Review · Reviewer_fYQg · 2024-11-04

**Soundness:** 2
**Presentation:** 2
**Contribution:** 2
**Rating:** 3
**Confidence:** 3

**Summary:**

This paper studies whether models can be flexible as to using the given context versus its parametric knowledge. In particular, they study whether models can handle 1) **context requests**: when the user asks the model to answer based on context, overriding any parametric knowledge, 2) **factual requests**: absent such an instruction, the model should answer based on its parametric knowledge, and 3) **NoAns requests**: when the context does not contain an answer, the model should say so.

To test this they create FaithfulBench, where context requests are created by synthetically revising passages accompanying factual questions, so that they support new, counterfactual answers. They find that no model is strong in all three categories. In general, models with more internal knowledge are less faithful to the contextual knowledge. Only instruction-tuning on data from one of the three categories will improve performance in that category, but at the detriment of others. Thus the authors propose BALCONI training, which basically means tuning the model on all three types of requests at once. They show that this achieves strong performance on both FaithfulBench and a new benchmark they design for context-based machine translation.

**Strengths:**

This paper formalizes three scenarios that models should excel at, in order to flexibly adapt to contextual vs. parametric knowledge. They show some of the tradeoffs that come with tuning the model with only one scenario in mind.

**Weaknesses:**

- There is a missing baseline: instead of tuning a model on each of the three request types (i.e., SFT-F, SFT-NoAns, and SFT-C baselines), which trades off with performance on the other request types, what if you give *in-context* examples corresponding to the desired request type? For instance, if the user wants the model to use context instead of parametric knowledge, then simply provide some in-context examples from the "context requests" category. This will likely teach the model in-context to rely solely on contextual knowledge, without compromising its ability to use parametric knowledge on other requests.
- The findings of the paper are generally unsurprising. Analysis of when models rely on contextual vs. parametric knowledge is well-studied, and BALCONI essentially boils down to training on all of the request types that the model will be evaluated on. Here is some related work about knowledge conflicts that aren't cited —

Rich Knowledge Sources Bring Complex Knowledge Conflicts: Recalibrating Models to Reflect Conflicting Evidence (https://arxiv.org/abs/2210.13701)
Knowledge Conflicts for LLMs: A Survey (https://arxiv.org/abs/2403.08319)
Resolving Knowledge Conflicts in Large Language Models (https://arxiv.org/abs/2310.00935)
Understanding the Interplay between Parametric and Contextual Knowledge for Large Language Models (https://arxiv.org/abs/2410.08414)

**Questions:**

- For TriviaQA, it looks like special care was taken to ensure the counterfactual passage makes sense and doesn't contain internal conflicts. Was this also done for Natural Questions?
- L. 196: the text states that three options are given to the human annotator, but I only see two listed (the factual one and the counterfactual one). What is the third option?
- What hyperparameter do you use for the context-aware decoding baseline?

---

> ### Author Response · Authors · 2024-11-20
> **Rebuttal to the reviewer**
>
> We greatly appreciate the careful comments and suggestions provided by the reviewer. Our response to the weakness is shown as follows:
>
> **W1**: what if you give in-context examples corresponding to the desired request type?
>
> **Answer:** In our dataset, the context length quite long, for example, the average number of the tokens is 382 in the TriviaQA and 338 in the NQ, which could cause the length out of the model max sequence length since that of mistral-7B is 2048. We carry out the ICL experiment in Qwen-2.5-7B-Inst, whose max sequence length is 128k, following your suggestion. In the TriviaQA multi-choice task, we observe that the ICL result in context request is 54.8, 1.7 lower than that of baseline (56.9), but 84.9 in NoAns request, 9.8 higher than that of baseline (74.7). The results may stem from the long context information for the model in the context request, but a simple pattern in the NoAns requests (the model merely needs to choose NoAns based on the ICL demonstrations). We have added such discussion in the Appendix 7.8.
>
> The prompt is that ‘Complete the instruction following the examples I show you. \n Example 1: {demonstration 1}. \n Example 2: {demonstration 2} \n Example 3: {demonstration 3} \n’
>
> **W2**: he findings of the paper are generally unsurprising. Analysis of when models rely on contextual vs. parametric knowledge is well-studied, and BALCONI essentially boils down to training on all of the request types that the model will be evaluated on. Here is some related work about knowledge conflicts that aren't cited.
>
> **Answer**:  Indeed, this study aims to highlight the finding that there is a negative correlation between the model's internal knowledge correctness and context faithfulness, even though the model has not been specifically trained on this type of data. Few studies have comprehensively analyzed this phenomenon and it is still an unsolved problem (not well-studied). To further substantiate this point, we conducted additional experiments using various model sizes of Qwen-2.5-Inst, which demonstrate this correlation. The results are presented as follows:
>
> # TriviaQA Multi-choice QA
>
> | Model                | Factual | Context | NoAns |
> |----------------------|---------|---------|-------|
> | Qwen-2.5-1.5B-Inst   | 45.4    | 70.6    | 39.3  |
> | Qwen-2.5-7B-Inst     | 67.1    | 56.9    | 74.7  |
> | Qwen-2.5-14B-Inst    | 71.7    | 36.9    | 94.7  |
>
> # NQ Multi-choice QA
>
> | Model                | Factual | Context | NoAns |
> |----------------------|---------|---------|-------|
> | Qwen-2.5-1.5B-Inst   | 38.0    | 49.9    | 87.4  |
> | Qwen-2.5-7B-Inst     | 49.1    | 43.4    | 99.4  |
> | Qwen-2.5-14B-Inst    | 55.2    | 31.3    | 99.2  |
>
> Additionally, we evaluated various fine-tuning strategies, which also revealed a negative correlation. In response, we propose a mixed training strategy to mitigate this issue. Existing research has primarily focused on how to respond to knowledge conflicts [1] or detect them [2]. We have included the references you mentioned in our related work section and highlight these differences more clearly.
>
> **Q1**: For TriviaQA, it looks like special care was taken to ensure the counterfactual passage makes sense and doesn't contain internal conflicts. Was this also done for Natural Questions?
>
> **Answer:** We construct the NQ and TriviaQA in different ways. We use the replacement method to process the NQ data since we expect to keep the same setting with previous studies such as [3-5], and as [6] stated the context generated by directly replacing the answer is difficult for LLM to be convincing. Thus we construct the TriviaQA part using LLMs. We have revised Section 3.1 to stress this point.
>
> **Q2**: L. 196: the text states that three options are given to the human annotator, but I only see two listed (the factual one and the counterfactual one). What is the third option?
>
> **Answer**: Sorry for the typo, the third option is I cannot tell the answer from the context, and almost no human answers fall in this option. We have elaborated the manuscript in the new version.
>
> **Q3**:  What hyperparameter do you use for the context-aware decoding baseline?
>
> **Answer**: We follow the published original script in the github repo context-aware-decoding for NaturalQuestion, where WEIGHT = 2_-1 and TOPP="0.0”. The setting of the hyperparameters is added to the Section 4.
>
> [1] Rich Knowledge Sources Bring Complex Knowledge Conflicts: Recalibrating Models to Reflect Conflicting Evidence
> [2] Resolving Knowledge Conflicts in Large Language Models
> [3] Ella Neeman, et al. DisentQA: Disentangling parametric and contextual knowledge with counterfactual question answering.
> [4] Wenxuan Zhou, et al. Context-faithful prompting for large language models.
> [5] Shayne Longpre et al. Entity-based knowledge conflicts in question answering.
> [6] Xie et al. Adaptive Chameleon or Stubborn Sloth: Revealing the Behavior of Large Language Models in Knowledge Conflicts.

---

### Official Review · Reviewer_9Kwx · 2024-11-05

**Soundness:** 3
**Presentation:** 2
**Contribution:** 3
**Rating:** 6
**Confidence:** 4

**Summary:**

This paper introduces FaithfulBench, a benchmark designed to evaluate the faithfulness of large language models (LLMs) and assess how improvements in faithfulness might impact internal knowledge accuracy within LLMs. Extensive experiments show that 1) LLMs exhibit unfaithful to the context to some extent; 2) there is a clear negative correlation between faithfulness and internal knowledge accuracy across different LLMs in multiple-choice QA tasks; 3) instruction tuning with counterfactual data can significantly improve the model's context faithfulness but compromise the model's internal knowledge. Based on "BALCONI training", training with mix-up data of factual requests, context requests, and NoAns requests can achieve a well-balanced effect in improving balanced faithfulness and internal knowledge.

**Strengths:**

1. This paper presents FaithfulBench, encompassing the tasks of Openbook QA and Multi-choice QA based on the NaturalQuestions and TriviaQA datasets. Experimental results show that all models exhibit a degree of stubbornness to their internal knowledge, and there exists a significant negative correlation between faithfulness and internal knowledge correctness in multi-choice QA.
2. This paper compares different previous methods in improving faithfulness and finds that t tuning the model on counterfactual data can enhance model faithfulness to the relevant context but hinder the model’s correctness in internal knowledge.
3. This paper proposes BALCONI training, tuning the model using mix-up data of factual requests, context requests, and NoAns requests, whose effectiveness has been proven in experiments.

**Weaknesses:**

1. While the paper presents a dataset focused on faithfulness, the data is generated via LLMs, which may introduce hallucinations during the dataset generation process and affect the evaluation results. It would be better if the authors could introduce some methods to guarantee the data quality and add human evaluation on (a subset of) data.
2. The models employed in the paper are somewhat outdated. For instance, Mistral 7B, which was introduced over a year ago, has since seen newer versions. Updating the experiments with the latest, more advanced models could potentially alter the results and observations.

**Questions:**

See "Weaknesses".

---

> ### Author Response · Authors · 2024-11-20
> **Rebuttal to the reviewer**
>
> We greatly appreciate the careful comments and suggestions provided by the reviewer. Our response to the weakness is shown as follows:
>
> **W1**: It would be better if the authors could introduce some methods to guarantee the data quality and add human evaluation on (a subset of) data.
>
> **Answer**: We have described the quality control and human evaluation in the section 3.1. To control the data quality, we check whether the counterfactual answer and the factual answer exist in the counterfactual context, and regenerate the counterfactual until the former exists but the latter doesn't. Given the counterfactual context and the question with three options the factual one, the counterfactual one and NoAns, the annotators are required to select the option using the contextual information. We then calculate accuracies of them, which are 93% and 91%, respectively. The results indicate the quality of the LLM generated data is satisfactory.
>
> **W2**: The models employed in the paper are somewhat outdated. For instance, Mistral 7B, which was introduced over a year ago, has since seen newer versions. Updating the experiments with the latest, more advanced models could potentially alter the results and observations.
>
> **Answer**: We conducted experiments using the recently released Qwen-2.5-Inst models, ranging from 1.5 billion to 14 billion parameters, on multi-choice QA tasks. We observed that as the model size increases, performance improves for factual requests and NoAns scenarios. However, there is a noticeable decline in performance on Context requests. This suggests a negative correlation between the correctness of internal knowledge and context faithfulness. We have added this experiment in the Appendix 7.7, which we will move to the main body in the final version.
>
> ## TriviaQA Multi-choice QA
>
> | Model                | Factual | Context | NoAns |
> |----------------------|---------|---------|-------|
> | Qwen-2.5-1.5B-Inst   | 45.4    | 70.6    | 39.3  |
> | Qwen-2.5-7B-Inst     | 67.1    | 56.9    | 74.7  |
> | Qwen-2.5-14B-Inst    | 71.7    | 36.9    | 94.7  |
>
> ## NQ Multi-choice QA
>
> | Model                | Factual | Context | NoAns |
> |----------------------|---------|---------|-------|
> | Qwen-2.5-1.5B-Inst   | 38.0    | 49.9    | 87.4  |
> | Qwen-2.5-7B-Inst     | 49.1    | 43.4    | 99.4  |
> | Qwen-2.5-14B-Inst    | 55.2    | 31.3    | 99.2  |

---

> > ### Comment · Reviewer_9Kwx · 2024-12-01
> > **Response to Authors**
> >
> > Thank you for your detailed response on data quality and experiments, which satisfies my questions on this paper. I have increased my scores accordingly and wish you all the best with your submission.

---

### Comment · Area_Chair_yytd · 2024-11-25
**Respond  to the author's rebuttals**

Dear Reviewers,

Thank you for your efforts in reviewing this paper. We strongly encourage you to review and respond to the author's comments to promote a more dynamic exchange of ideas.

Thank you for your collaboration.


Best regards,

ICLR 2025 Area Chair

---

### Meta-Review · Area_Chair_yytd · 2024-12-19

**Metareview:**

The paper proposes FaithfulBench, a benchmark used to evaluate the faithfulness of large language models (LLMs) in balancing reliance on contextual information versus their internal knowledge. The study investigates how LLMs handle different types of requests: context requests, factual requests, and NoAns requests. The findings indicate that LLMs often struggle to maintain faithfulness to context while utilizing their internal knowledge accurately. The authors also propose the BALCONI training method, which involves fine-tuning models with a mix of factual, context, and NoAns requests. This approach aims to enhance the models' ability to appropriately use context without compromising internal knowledge accuracy.

However, the title of the paper emphasize the proposed method, BALCONI, while the structure of the paper predominantly centers around the benchmark, FaithfulBench. Towards the end of the paper, Section 5.5 presents a non-standard machine translation task to demonstrate the versatility of the proposed method. This makes the paper not clear and difficult to follow.

**Additional Comments On Reviewer Discussion:**

1) The novelty of the paper
2) The quality of the proposed FaithfulBench
3) The baseline model, Mistral 7B, is somewhat outdated
4) The details regarding the experiments

The authors' responses have effectively addressed the reviewers' concerns.

---

### Decision · Program_Chairs · 2025-01-22

Reject